# Species responses to weather anomalies depend on local adaptation and range position

Yolanda Melero [1,2,3] ✉, Luke C. Evans[2,4], Mikko Kuussaari[5], Reto Schmucki [6], Constantí Stefanescu[7], David B. Roy [6,8] & Tom H. Oliver [2]

Species show intra-specific variation in responses to climate change linked to adaptation to the local climatic conditions. Likewise, species are expected to be more resilient to climate change at the centre of their bioclimatic niche, but this pattern is not general. We show that species sensitivity to climatic anomalies varies with local adaptation and the position in the bioclimatic niche, using long-term butterfly monitoring data for 34 species. Climatic anomalies negatively affected all populations of locally adapted species. Globally adapted species were positively or negatively affected by climatic anomalies, depending on population location and direction of anomalies. These responses impacted population trends as globally adapted species showed steeper declines at the trailing margin. Surprisingly, locally adapted species showed stable abundances at the trailing margin, but declines at the leading; which could be explained by the with the 'warmer is better' hypothesis where thermodynamics limit insect performance at cooler conditions.

Species sensitivity to climate change, especially in the form of extreme climatic events, plays an important role in explaining variation in population dynamics[1-3]. In some cases, populations across the species' entire distribution can fluctuate synchronously, while for other species, population dynamics are more independent[4-6]. Species adaptations to the local biotic and abiotic conditions, due to species phenotypic plasticity or evolutionary change of their traits across populations, may partly explain this observed intraspecific variation in population change[7,8]; but the interaction between species adaptations and range position is not simple to resolve.

Large-scale population studies show distinct or even opposite populations responses to large-scale climatic anomalies in relation to their position in the distributional range, especially for populations in the marginal edges of their range[9-11]. For these species, populations at the centre of the distributional range usually show higher population stability than those at the edges (*cf.* the 'abundant centre hypothesis'[12-14]); while populations at the distribution margins tend to be more sensitive to environmental variations and, hence, more vulnerable and unstable[7,15]. This pattern is expected because, although some environmental (e.g., altitudinal differences[16]) and methodological[17,18] nuances apply, there is a general concordance between the geographical/distributional range space and the species' optimal

environmental space (the bioclimatic/ecological niche, but see[19])[6,13,20,21]. Hence, populations at the distribution range centre are closer to the centre of the bioclimatic niche and more stable and abundant as a consequence (i.e., the 'abundant niche centre hypothesis'[20]), while populations at the distribution margins are nearer to the thresholds of the species tolerances (e.g., minima or maxima of thermal performance)[15]. In this paper, we adopt this bioclimatic niche approach and define leading and trailing range margins with respect those these bioclimatic niche boundaries (see Methods).

A general expectation then is that population dynamics at different edges of the bioclimatic niche are expected to be independent or contrasting in response to large-scale climatic anomalies. In fact, when climatic anomalies occur, the population performing best across the range might not be the central population, as the anomaly creates conditions away from the performance optimum at the centre and towards the optimum at one of the range margins. For example, increases in temperatures can facilitate increased population growth at sites where the temperature is typically below the optimum for that particular species (termed 'leading margin' hereafter) but may lead to population reductions at the centre and at sites with average temperatures above the optimum ('trailing margin' hereafter)[22].

[1]Department of Evolutionary Biology, Ecology and Environmental Sciences & Biodiversity Research Institute (IRBio), Universitat de Barcelona, Barcelona, Spain. [2]School of Biological Sciences, University of Reading, Whiteknights, Reading, Berkshire, UK. [3]CREAF, Bellaterra, Cerdanyola del Vallès, Spain. [4]Butterfly Conservation, Wareham, Dorset, UK. [5]Finnish Environment Institute (SYKE), Nature Solutions Unit, Helsinki, Finland. [6]UK Centre for Ecology & Hydrology, Wallingford, Oxfordshire, UK. [7]Natural Science Museum of Granollers, Granollers, Spain. [8]Centre for Ecology and Conservation, University of Exeter, Penryn, UK. ✉ e-mail: ymelero@ub.edu

Under climate warming, many species are shifting their distributions with population increases and expansions at the leading margins and local extinctions and range contractions at the trailing margins[23–26]. This pattern is expected only if populations share the same response to anomalies across their bioclimatic niche (i.e., same performance curves across populations; here termed 'globally adapted species' since they do not show adaptations to the local climatic regime, see Supplementary Note). However, this pattern would be disrupted if populations are locally adapted to the climatic conditions of their site (i.e., species whose populations show different performance curves, each adapted to the local regime; named 'locally adapted species', see Supplementary Note). For these species, their dynamics are expected to be more synchronised as large anomalies are expected to cause relatively similar deviations from the local climatic conditions.

The pattern and strength of responses along a species' bioclimatic niche axis (and corresponding distributional range) may be hence modulated by the degree of adaptation to the local conditions. While this 'degree of local adaptation' is difficult to capture, here we build on previous research that develops a simple measure of local (physiological) adaptation by whether a species responds more strongly to local climatic anomalies or global climate anomalies (measured through $R^2$; see Supplementary Note). This simple surrogate for local adaptation allows us to make some general predictions about the interaction between species adaptations and distributional range position.

We hypothesized that the direction and shape of species responses to climatic anomalies will vary along the species bioclimatic niche, due to their degree of local adaptation. We assessed species' bioclimate niches through analysis of species-specific temperature and precipitation variables, and used a previous methodology to assess the species degree of local adaptation[8], in order to differentiate locally versus globally adapted species (see Supplementary Note). Specifically, we predict that population growth rates of locally adapted species follow a non-linear quadratic response to local climatic anomalies, with a maximum performance around the local average conditions (i.e., when no anomalies occur) decreasing below and above them (i.e., when local anomalies occur). We also expected the population changes of locally adapted species to be independent of the position within the species bioclimatic niche. However, for globally adapted species we predicted population responses to the local climatic anomalies to differ depending on the population position along the species bioclimatic niche. For these species, we predicted a non-linear quadratic response for the populations located at the centre of their bioclimatic niche (i.e., at the location of their optimal conditions), but expected broadly linear responses at both their leading and trailing margins, e.g., anomalies increasing the local average temperature would lead to a broadly linear increase of population growth rate in the leading margin while a linear decrease in the trailing margin (Fig. 1).

Consequently, we also predicted species population trends, in terms of abundance over time, to differ along the species bioclimatic niche and between locally and globally adapted species. Specifically, we expected population trends of locally adapted species to be similar along their niche and more stable than those of globally adapted species. However, for the latter, we expected their abundances to show a reduction over time at their trailing margin and at the centre of their niche, but an increase over time at their leading margin.

To test our hypotheses, we performed a multi-species analysis using 34 butterflies, previously classified as locally or globally adapted species based on their degree of local adaptation (see Supplementary Note)[8], as a study system (Supplementary Figs. 1–34). We chose butterflies because they respond rapidly to environmental changes[27] minimizing the demographic time lag to extinction compared to e.g., plants and birds[28,29]. They also show a continuum across species in their degree of local adaptation to climatic conditions related to phylogeny[8]. Moreover, butterflies, like many other thermophilus organisms, are highly sensitive to weather and strongly affected by climate change[30–32], with best performances at certain optimum weather conditions but rapidly declining away from them[33]; as such, their distributional ranges strongly relate to their bioclimatic niches[7]. Lastly,

butterflies offer spatially- and temporally-replicated standardized abundance sampling data across a large temporal and continental extent, making them ideal to exploit 'natural experiments' and to test hypotheses about responses to temperature in real-world conditions.

## Results and discussion
### Population change, degree of adaptation and niche position

We use count data from 21 locally and 13 globally adapted species, classified previously by their degree of local adaptation (dla)[8] (dla =]0, 1] and dla = [-1, 0[, respectively; Supplementary Figs. 1–34). Count data were collected weekly as part of one of three European butterfly-monitoring schemes across 813 sites within six European bioclimatic regions between 1999 and 2017 (97,664 site-year-species data points). Data were used to calculate a series of annual abundance indices per species and site[34]. To test our first hypothesis, we modelled how annual population change varied in relation to the local climatic anomalies in interaction with the bioclimatic position of each population while accounting for density dependence given its importance in butterfly dynamics[35,36]. Models were fitted for locally and globally adapted species separately.

Climatic anomalies were calculated for the weather variable most affecting each species (temperature, precipitation or aridity) at the most sensitive species-specific phenological period (overwinter, pre-flight, flight, or post-flight period) and time (year t excluding post-flight period, or year t – 1; Supplementary Table 1)[7,8]. The bioclimatic niche of each species was constructed based on its distribution along the values of the climatic variable associated with the corresponding species; i.e., the minimum-to-maximum climatic values where the species has been detected during the study period along all studied sites; ranking −1 (leading margin) to 1 (trailing margin; Supplementary Fig. 35).

We found species population sensitivity to the local climatic anomalies varied both depending on whether they were locally or globally adapted species and, for the latter, on the position of the population within the species bioclimatic niche (Fig. 2). The patterns of their responses were maintained even when taking a more conservative approach by removing potential outliers and selecting species with conservative values for degree of local adaptation (Supplementary Fig. 36 and Supplementary Table 2).

Populations of locally adapted species showed a unimodal 'n-shaped' response as predicted (model $R^2_m = 0.24$, $R^2_c = 0.36$), with a maximum population performance (highest population increase) around the mean conditions (i.e., when no local anomalies occurred) that declined towards the extremes of climatic anomalies, though peak performance was for anomalies slightly below the mean conditions (Fig. 2a). This n-shaped response was common for all locally adapted species independently of their position within their bioclimatic niche; but their average performance improved as we moved from the leading to the trailing margin (Fig. 2a and Supplementary Table 2; illustrative examples provided for *Brenthis ino* and *Satyrium esculi*, with the latter showing slightly better performance above and below the mean conditions for the leading and the trailing margins respectively; Fig. 3a, b and Supplementary Table 3).

Matching our predictions, populations of globally adapted species showed broadly linear responses to the local climatic anomalies (model $R^2_m = 0.24$, $R^2_c = 0.50$; Fig. 2b and Supplementary Table 2; illustrative examples provided for *Satyrium spini* and *Ochlodes sylvanus*, Fig. 3c, d and Supplementary Table 3). Climatic anomalies had close to null effect on the populations located at the centre of their species bioclimatic niche (Fig. 2b), indicating resilience to a wide range of conditions in concordance with their global adaptation[8] and also possibly as a result of higher genetic variation at the species bioclimatic (ecological) niche centre[37] increasing population stability. Populations at the trailing margin declined with the increase of the climatic variable (indicating hotter and drier years) and performed best during negative anomalies (cooler and wetter years). Inversely, populations at the leading margin increased with the increase of the climatic variables (hotter and drier years), but performed worst with negative climatic anomalies (cooler and wetter years). This indicated that the population performance of globally adapted species decreased or improved as climatic

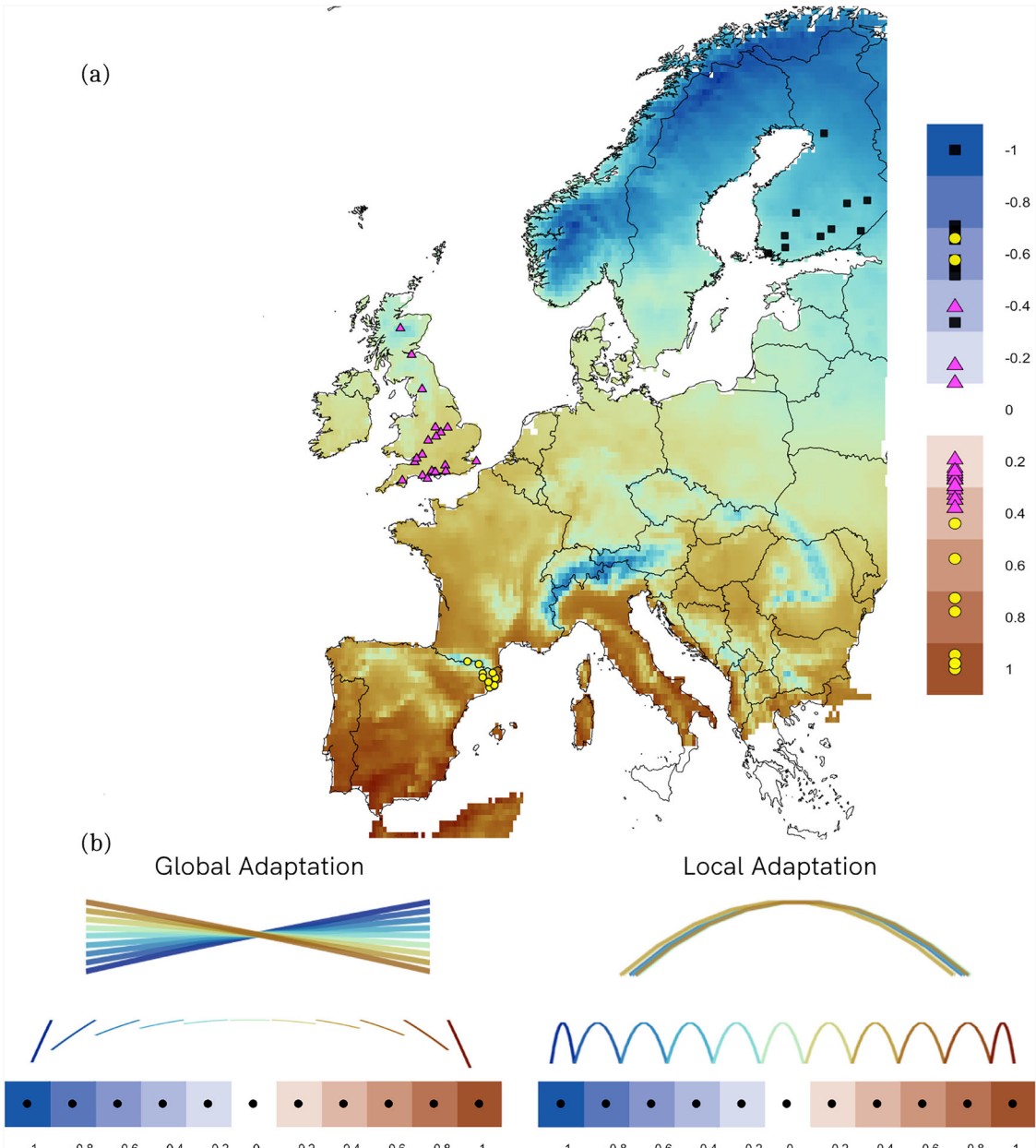

**Fig. 1 | Simulated predictions of population responses of globally and locally adapted species to local climatic anomalies in relation to population niche position. a** shows a random subset of the study sites across the three countries (dots representing sites coloured and shaped by country: yellow circles - Spain, magenta triangles—UK, and black squared—Finland) representing a hypothetical species distribution for one thermal generalist species with populations with populations spanning the bioclimatic niche (see Methods) from the leading edge (niche position = −1) to the trailing margin (niche position = 1). The background colour of the map reflects mean annual temperature across the study period (with brown showing warmer conditions). **b** shows predicted responses to local climatic anomalies for populations spread across the full bioclimatic niche space for species that are either globally or locally adapted, with warmer sites shown in deeper brown colours. The local response in panel b shows performance (i.e., population change) in response to increasing anomalies (low to high) at each site (derived from a model fit simulating expected population growth in relation to the interaction between bioclimatic niche position and local climatic anomaly). The simulation code is available at [https://doi.org/10.5281/zenodo.15065537].

conditions either moved away- or approached- the species central optimal conditions, respectively[8]. For example, populations at the trailing margin of *Ochlodes sylvanus* performed worst in years of higher temperature relative to the annual local mean (Fig. 3c), but these declines became less steep towards the centre of the species bioclimatic niche, and shifted to positive responses towards the leading margin, i.e., populations at the leading margin were favoured by hotter years (Fig. 3d).

Overall, locally adapted species showed an optimum (close to) the local mean conditions and globally adapted species showed the most consistent high performance at the centre of the species bioclimatic niche. Hence, our

results support the abundant niche centre hypothesis[12,13,20,38] for globally adapted species, since species optimal conditions coincided with the centre of their bioclimatic niche. But, for locally adapted species, the optimal (central niche) conditions were centred on the average local climatic regime, rather than at the centre of the species entire bioclimatic niche. The observed differing responses of species to climatic conditions in relation to their local or global adaptation provide new insights for the contrasting (or lack of) results linking variation in species abundances along their distributional range and ecological (bioclimatic) niche[20,39] and their responses to climate change[7,14]. Further, while our results were consistent for both categories

**Fig. 2 | Population change in relation to the local climatic anomalies of the year and the population position within the species bioclimatic niche. a, b** show model predictions for locally ($N = 21$, $n = 37,553$) and globally ($N = 13$, $n = 27332$) adapted species, respectively. Divergent responses according to the position of the site in the species bioclimatic niche are shown at 0.1 intervals from the leading (niche position $= -1$) to the trailing margin (niche position $= 1$), displayed from leading to trailing (blue and red, respectively; white indicates centre).

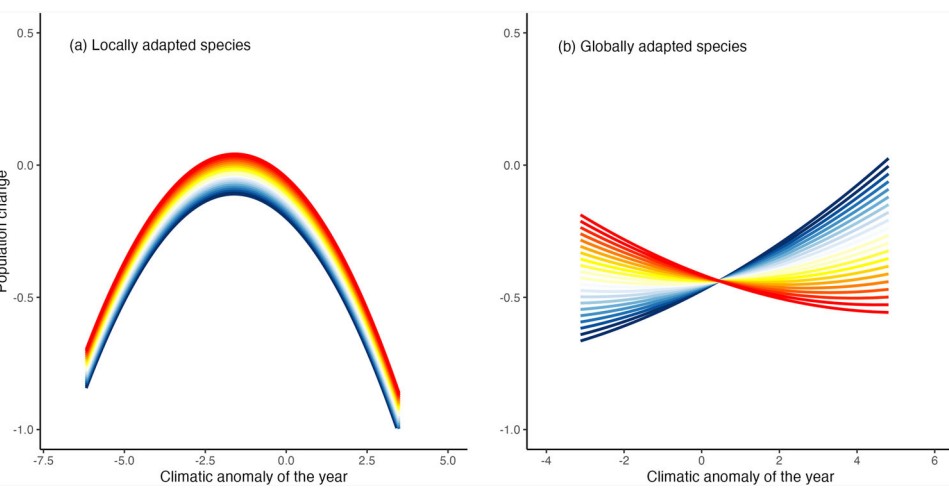

(locally and globally adapted) independently of the degree of local adaptation used for their categorisation (i.e., sensitivity analysis based on either all or only more conservative values and without potential outliers; Supplementary Fig. 36 and Supplementary Table 2), which explains slight variations in the responses of species within the same categories (e.g., Fig. 3a, b and Fig. 3c, d).

We also carried out a version of the analysis which controlled for phylogenetic non-independence (since we found previously a phylogenetic signal in degree of local versus global adaptation[8]). Results were qualitatively similar can be found in Supplementary Table 4.

### Abundance over time of locally and globally adapted species

To test our second hypothesis on population trends of locally adapted species being similar and stable along their niche, while populations of globally adapted species showing reductions and increases at their trailing and leading margins respectively, we modelled the annual population abundance in relation to the cumulative number of years of observations per species in an interaction with the bioclimatic niche position of each population. As per our hypotheses and above results, we fitted the models to locally and globally adapted species separately. We found considerable variation in species population abundances over time for both locally and globally adapted species (p-values $< 0.0001$ for both species categories; models $R^2_m = 0.008$, $R^2_c = 0.44$ and $R^2_m = 0.002$, $R^2_c = 0.51$, respectively, with species nested with the timeseries, and sites, set as random effects; Supplementary Figs. 37–70 and Supplementary Table 5). This is expected given the multiple factors that simultaneously affect annual population dynamics beyond key weather variables and the sampling procedure. We also found a significant variation across the species bioclimatic niche, with opposite directions for locally and globally adapted species (Fig. 4 and Supplementary Table 5).

Despite the observed variation among species, we detected a general decline of the abundances of locally adapted species over time. Contrary to our predictions, declines varied with bioclimatic niche position and the decline was particularly steep at the leading margin but smoothed towards the centre, shifting to a slight increase of abundances for those populations located at the trailing margin (Fig. 4a). This pattern was consistent when taking the conservative approach of removing potential outliers and selecting species with conservative values for degree of local adaptation (Supplementary Fig. 71a–c and Supplementary Table 5).

Populations of globally adapted species also showed a generalized decline of abundance over time for all populations. However, and inversely to populations of locally adapted species, this decline was steeper at the trailing margin while close to null at the leading margin of the species bioclimatic niche (Fig. 4b). This pattern was also robust when removing outliers and uncertainty in the degree of local adaption; though the decline of populations at the trailing margin was less steep and populations at the

leading margin slightly increased in abundances when removing potential outliers (*Laeosopis roboris* with dla $= -0.22$, and *Cupido osiris* with dla $= -0.23$; Supplementary Fig. 71d, e and Supplementary Table 5).

Our results indicate that leading margins are a stronghold of species persistence under climate warming for globally adapted species, but not for those adapted locally.

### Implications for forecasting and conservation

Our results reveal that population responses to climate change are strongly contextualised by the species' ability to adapt to the local climatic conditions[8] and the location of the population within the species bioclimatic niche. Our results challenge the generally accepted pattern of climatic change leading to increased population abundance at the leading margins while decreasing the abundance of populations at the trailing margins. The responses of locally adapted species to climatic anomalies were independent of their location, but we found that populations at the leading margin had lower average population growth and, hence, greater long-term declines in their abundances. The responses of globally adapted species, however, were dependent on their location at the leading, centre or trailing margin, and population trends were consistent with the expected effects of climate change with greater declines at the trailing edge of the bioclimatic niche.

Our results are largely consistent with expectations from theory and previous empirical studies of thermal performance. We found that locally adapted species suffered from steep reductions in performance away from the average local climatic conditions (e.g., all populations of a locally adapted species will suffer reductions during colder and wetter years, and during hotter and drier years), whereas responses to local climatic anomalies were less severe for globally adapted species and the direction of performance change depended on the location of the population (e.g., hotter and drier years benefit populations at the leading margin but reduce those at the trailing). This pattern is consistent with trade-offs in thermal specialism-generalism, where for thermal specialists a higher performance at the optimum is traded off against steep reductions in performance away from the optimum[40], whereas thermal generalists sacrifice lower performance at the optimum for more consistent performance across a range of temperatures. The consistency between our results and previous theory also validates our simple approach (the local degree of adaptation; see Supplementary Note) to capture local physiological adaptations.

We also found for locally adapted species that average population growth was lower at the leading margin. This pattern aligns with the 'warmer is better' hypothesis[41] where, in insects, adaptation to cooler conditions can improve performance to some extent but performance is ultimately limited by thermodynamic constraints such that populations adapted to warmer optimum temperatures will outperform those adapted to cooler temperatures at their optimum temperatures respectively[42].

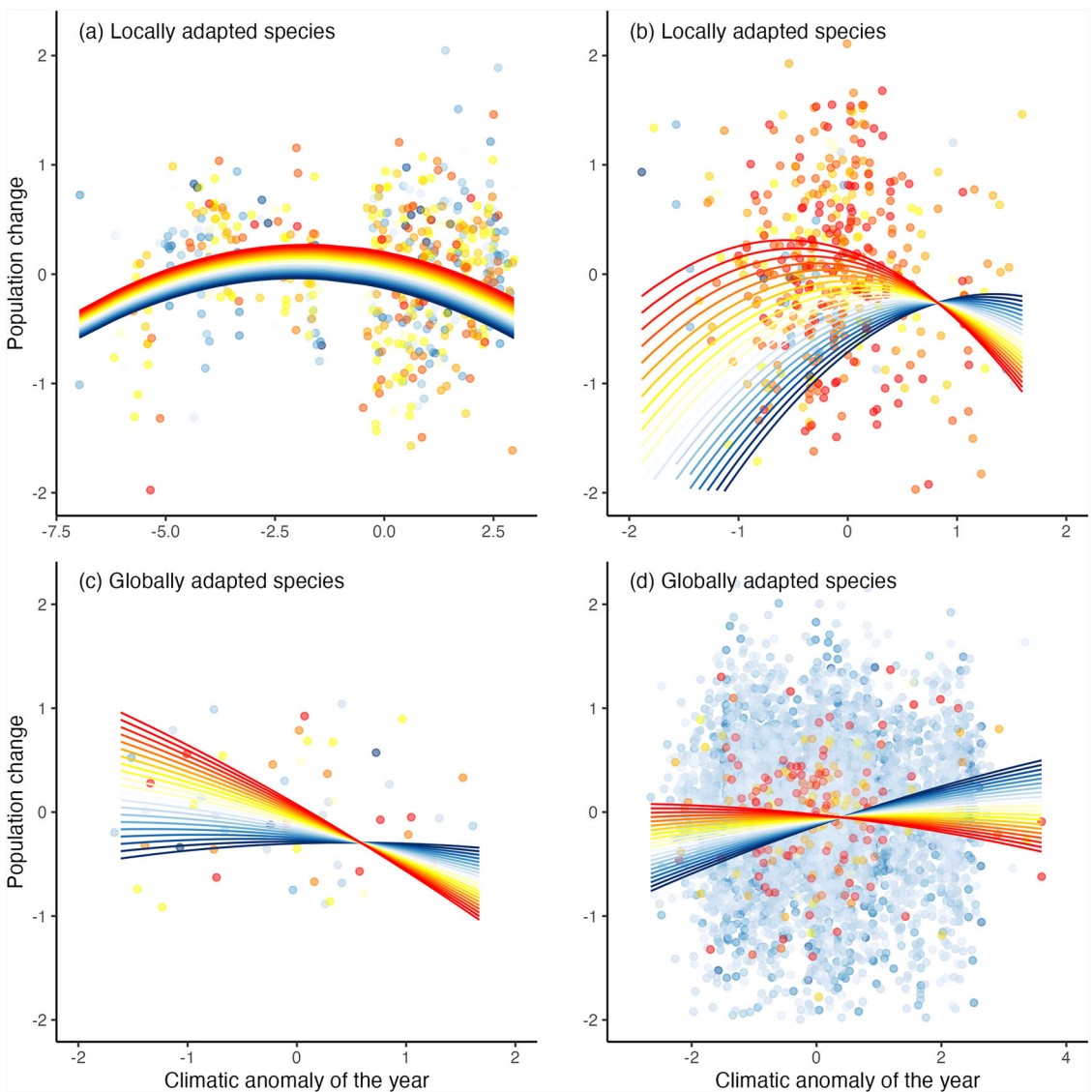

**Fig. 3 | Illustrative specific examples of population change in relation to the local climatic anomalies of the year and the population position within the species bioclimatic niche.** Panels show residuals (dots) and model predictions of population change for the locally adapted species in (**a**) *Brenthis ino* (n = 1194; degree of local adaptation = 0.06) and (**b**) *Satyrium esculi* (n = 878; degree of local adaptation = 0.05), and for the globally adapted species in (**c**) *Satyrium spini* (n = 112; degree of local adaptation = −0.02) and (**d**) *Ochlodes sylvanus* (n = 12794; degree of local adaptation = -0.04), in response to the local annual anomaly of the climatic variable most affecting each species (temperature for (**a**, **b** and **d**); and, shown in inverse, precipitation for (**c**)). Divergent responses according to the position of the site in the species bioclimatic niche are shown at 0.1 intervals from the leading (niche position = −1) to the trailing margin (niche position = 1), displayed from leading to trailing (blue and red, respectively; white indicates centre). Dots colours also relate to the position of the site in the species bioclimatic niche.

Additionally, for locally adapted species we found that the optimum performance was below the average mean conditions of the site, a strategy found to be optimal for maximising performance in variable environments due to the asymmetry in the rate of performance decline above and below optimum conditions[43,44].

A limitation of the data is that, due to sampling locations, we do not evaluate data from the true trailing margin of some species (i.e., the trailing margin is further south in hotter and drier conditions, e.g., *Melanargia galathea* distribution extends to North Africa, *Leptidea sinapis* to Southern Spain; Supplementary Table 1). Consequently, we may see poor performance at the leading margin due to the aforementioned limits on adaptation in cooler conditions, but we do not observe the populations experiencing the most detrimental effects of increased temperature and aridity.

In conclusion, we have shown that incorporating information on locally adapted species and for globally adapted species their location is likely to be critical to reduce future population extirpations and species loss, and, in consequence, for developing effective adaptation measures for biodiversity[45]. Specifically, locally adapted species can benefit from management actions buffering the displacement of their local optimal conditions when anomalies occur, while globally adapted species can benefit from contextualised management actions as per their location and the direction of the site anomalies (i.e., buffering against cold wet anomalies at the leading margin but hot dry ones at the trailing edge). For example, locally adapted species may need more variety of microhabitats to help them thermoregulate when the weather disrupts the local conditions. Due to their local adaptation this is important across the distributional range, though seems especially important at the leading margin, where populations of locally adapted species show steeper declines; while globally adapted species may need more cooler, damper microsites such as scrub and woodland edges to survive high temperature and aridity anomalies at trailing margins.

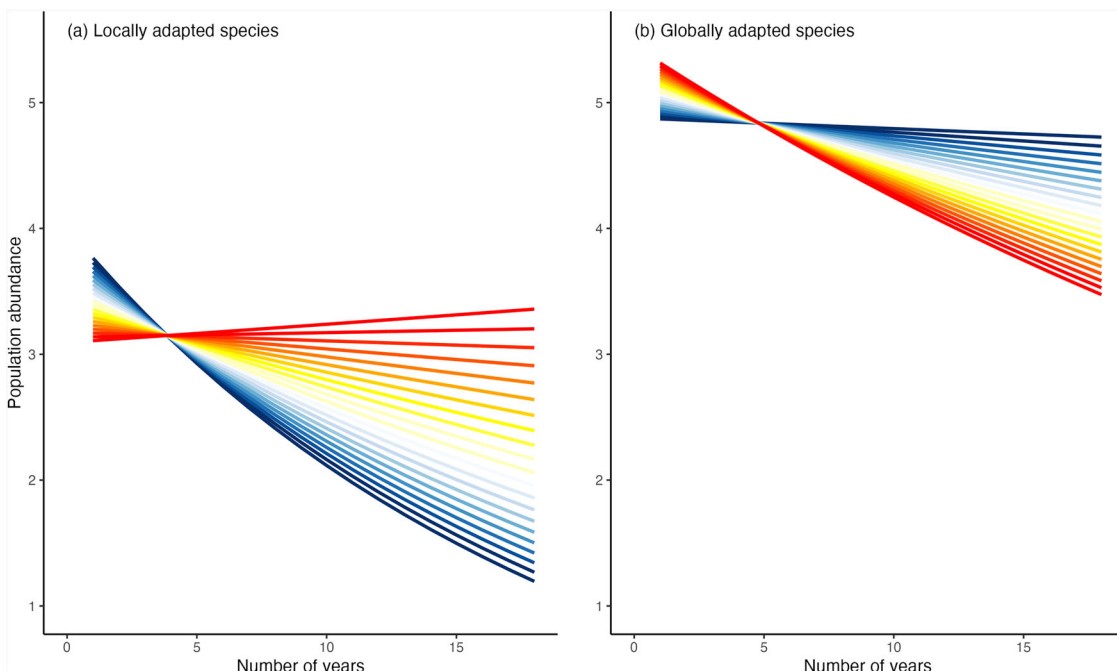

**Fig. 4 | Population abundances over time in relation to the population position within the species bioclimatic niche. a, b** show model predictions for locally ($N = 21$, $n = 50718$) and globally ($N = 13$, $n = 41692$) adapted species, respectively. Divergent trends according to the position of the site in the species bioclimatic niche are shown at 0.1 intervals from the leading (niche position = −1) to the trailing margin (niche position = 1), displayed from leading to trailing (blue and red, respectively; white indicates centre).

## Methods

### Degree of local adaptation

We used count data from 34 butterfly species whose populations have been previously seen to show a clear response to specific climatic anomalies[8] (Supplementary Figs. 1–34). These species have been previously classified as locally ($N = 21$) or globally ($N = 13$) adapted to the climatic regime based on their degree of local adaptation[8]. The latter is a continuum value measuring the species-specific sensitivity to climatic anomalies at two spatial scales: local (i.e., climatic deviations from the average across all years at that specific site) and global (i.e., climatic deviations from the average climate across all years and all sites where the species is found); quantified as the difference in the variance explained by the models:

$$\log\left(\frac{N_{it}}{N_{it-1}}\right) = N_{it-1} + W_{i,local\,t'} + W_{i,local\,t'}^2 + \varepsilon \quad (1)$$

$$\log(N_{it}/N_{it-1}) = N_{it-1} + W_{i,global\,t'} + W_{i,global\,t'}^2 + \varepsilon \quad (2)$$

where $N_{it}$ is the annual abundance index calculated per site ($i$) and time (year, $t$); and $W_{it}$ is the standardized climatic anomaly of the weather variable and for the temporal window ($t'$) most affecting the dynamics of the species at the site ($i$), either at the local or the global scale[8]. We included a density dependence term ($\log N_{ijt-1}$) because it is important in butterfly populations[35,46], as well as the quadratic slope coefficient of the climatic anomaly to detect potential for non-linear responses. Site was set as a random intercept, except for species with a number of sites <10, and errors were set as normally distributed after verification.

The climatic variable $W_{it'}$ was calculated for temperature, precipitation and aridity (as a combination of temperature and precipitation) since butterflies, and many other organisms, can be affected by each of these variables[7,8,30,47]. Likewise, the temporal window ($t'$) was defined for each life stage, given their relevance at modulating the effect of climate on population dynamics for many organisms, including butterflies[48]. These were: pre-flight, flight, post-flight or overwintering period[7] and year $t$ and $t-1$ (to account for delayed responses), set per species and bioclimatic zone. Each

period was defined using the annual flight curve distribution from the relative abundances per species and zone (i.e., using a generalised additive model fitted annual species phenology[34,49]) as follows: flight period was set as the dates between the 10th and 90th percentiles of a species abundance distribution per year and bioclimatic zone for both univoltine and multi-voltine species (i.e., species with more than one reproduction per year); pre-flight period was then set from February to the 10th percentile of the species flight period; post-flight period from the 90th percentile to the end of October for year $t-1$ (year $t$ was discarded because adults would be perished, so they have not possible effect on the counts or on the population dynamics of the species in that year); and overwintering period fixed from November to January (of year $t-1$ to year $t$)[7,8].

The most sensitive scale of adaptation (local or global), weather variable and temporal window (phenological period and year $t$ or $t-1$; Supplementary Table 1) per species were then selected based on AIC best model fit in relation to population growth rate (Eq. 1 and Eq. 2). Comparison with null model accounting only for density dependence was previously tested[8].

The degree of local adaptation was then quantified as the difference between the variance explained ($R^2$) by the best model in relation to the anomalies at the local scale versus the variance explained by the model at the global scale[8]. Hence, values of the degree of local adaptation range [−1, 1] from most sensitive to anomalies at the entire species distributional scale (globally adapted species) to most sensitive to those anomalies at the local (site) scale (locally adapted species). The degree of local adaptation of the selected 34 species ranged from [0.004, 0.7] for local, [0.004, 0.06] if removing the outlier *Parnassius apollo*, with median 0.03 for both cases; and [−0.23, −0.002] for global species, [−0.07, −0.002] if removing the potential outliers *Laeosopis roboris* and *Cupido osiris*, median −0.02 for both cases (Supplementary Fig. 72 and Supplementary Table 1).

### Data gathering

Species count data were collected via the long-term Butterfly Monitoring Schemes carried out in Finland, Spain and the UK covering six out of the ten bioclimatic regions across Europe (Supplementary Fig. 73). The schemes consist of a network of sites where volunteers perform weekly counts of

butterflies along a set of transects following the standardized 'Pollard Walk' methodology (Pollard, 1977). Monitoring is done during the butterfly flight season, which varies depending on the climatic zone within the range of beginning of March to end of September. The three schemes differ in starting year and number of surveyed sites: Finland (1999, $n_{sites} = 107$), Spain (1994, $n_{sites} = 130$) and the U.K. (1976, $n_{sites} = 2128$). Therefore, we used Finland as the limiting country to set the range of study years (1999–2017). To have sufficient data per site, we only used data from those transects with at least ten years of interannual population change between 1999–2017, leading to 53 transects in Finland, 59 in Spain and 701 in UK. Counts were transformed to an index of abundance ($N_{jit}$) per species, site and year using generalized additive models that account for missing counts, and spatial and temporal variation in the species phenology[34,49]. However, to assure consistent estimates, we excluded indices of abundance with more that 50% of weeks missing data. We used this annual species index of abundance to calculate the annual population change as $\log(N_{jit}/N_{jit-1})$.

We extracted the site daily temperature (in degrees Celsius) and precipitation (in millimetres) at 0.1° spatial scale (~11 km) for each site from the European Climatic and Assessment Dataset project (ECAD)[50,51], to then calculate the climatic anomalies and to define the bioclimatic niche of the species (see Bioclimatic niche construction). Potential interactions between temperature and rainfall were accounted for by calculating the standardized aridity index[47,52]:

$$SAI_{it} = -((P_{it} - P_i)/sd\,P_i)) * 0.5(T_{it} - T_i)/sd\,T_i); \tag{3}$$

where $SAI$ stands for Standardized Aridity Index (hereafter aridity), $T$ for mean temperature, $P$ for total precipitation, $sd$ for standard deviation calculated per site ($i$) and year ($t$).

For each species, we calculated the site climatic anomalies as:

$$W_{it'} = \bar{W}_{it'} - \bar{W}_i; \tag{4}$$

where $W_{it'}$ is the anomaly, $\bar{W}_{it'}$ and $\bar{W}_i$ are the mean temperature, the total precipitation or the aridity index (SAI) over the time series, depending on the climatic variable to which each species was most sensitive to, calculated per site ($i$) and, for $\bar{W}_{it'}$, per temporal window ($t'$; phenological period and year $t$ or $t$-1) during which the climatic anomalies most affected the population dynamics of the species (Supplementary Table 1)[8].

## Bioclimatic niche construction
We defined the species-specific bioclimatic niche following the realised niche concept, which delimitates the range in which a species can persist based on the physical environmental conditions in combination with biotic constraints (Hutchinson, 1957). For each species, the bioclimatic niche was constructed based on the species distribution along the minimum-to-maximum climatic values where the species was detected during the study period along all studied sites; using, per species, the climatic variable most affecting its population responses (temperature, precipitation or aridity; Supplementary Table 1).

Each species bioclimatic niche was calculated taking the annual mean climatic value of the temporal window associated to the species, at the site where the species was recorded as:

$$2[W_{it'} - \min(W)/\max(W) - \min(W)] - 1, \tag{5}$$

Where $W_{it'}$ is the mean climatic value per site ($i$) and temporal window ($t'$) and $W$ is the observations of climatic values from all sites and years[15]. The bioclimatic niche for precipitation was calculated with its inverse values, so that the bioclimatic niche was coincident with the species distribution along the temperature and the aridity index (e.g., coldest marginal edge will also have higher levels of precipitation).

This bioclimatic niche calculation preserves the relative difference between the climatic values of the sites where a species was observed but standardises the climatic range differences between species. As a result, the

minimum mean climatic value each species was ever detected was given a score of −1 and the maximum of 1. Thus, every site and species combination had a position within a range between −1 and 1, delineating the bioclimate leading and trailing margins, respectively and broadly corresponding with the edges of the species distribution (Supplementary Fig. 35).

To assure the alignment of the bioclimatic niche and the species population range position, we performed species specific Pearson correlations between them (Supplementary Fig. 74). To do so we used the relative range position, defined as the relative position of each population within the latitudinal distribution of its species and calculated based on the formula[7]:

$$RRP_i = (\bar{L}_i - \min(L_i))/(\max(L_i) - \min(L_i)) \tag{6}$$

where $L_i$ is the latitude, being $\bar{L}_i$ the average latitude of the species latitudinal distribution set as a vector, weighted by the number of data points (species abundance) from each site. $RPP$ is then expressed as a proportional range position standardised from 0–1 for each species, with 0 zero relating for lower latitudinal position (hence trailing marginal edge of the species) and one for higher latitudinal position (hence leading marginal edge of the species). While we lack of the entire distributional information for each species, this method allowed us to test for a proxy of correlations in latitudinal gradients, hence accounting for trailing and marginal edge margins.

## Statistics and reproducibility
We fitted linear mixed models to test if the variation of the species populations responses, in terms of population change, to the climatic anomalies along their bioclimatic niche differed between globally and locally adapted species. We set population change ($\log(N_{jit}/N_{jit-1})$) as the response variable, and fitted local climatic anomaly (at the site level) in an interaction with the position of the site within the species bioclimatic niche. We also included density dependence ($\log N_{jit-1}$) and the quadratic slope coefficient of the climatic anomaly to account for possible non-linear responses for each species. Species and site were set as random effects, and the error as normally distributed:

$$\log\left(N_{jit}/N_{jit-1}\right) = \log(N_{it-1}) + W_{it'}x\,B_{ji} + W_{it'}^2 + \varepsilon, \tag{7}$$

where $N_{jit}$ is the annual abundance index of the species $j$ at the site $i$ and year $t$, $W_{it'}$ is the climatic anomaly variable at the site $i$ and temporal window of the species (phenological period and year $t$ or $t$-1; Supplementary Table 1) at the local scale, and $B_{ji}$ is the bioclimatic position of the population of species $j$ at the site $i$. Models were fitted with the species most affecting standardised climatic variable $W$ (temperature, precipitation and aridity), and temporal window[8]. Climatic anomalies were standardised separately per variable (temperature, precipitation and aridity), and precipitation was used in inverse (i.e., multiplied by −1). Models were fitted pooling all species together but separately for locally (degree of local adaptation] 0, 1]) and globally (degree of local adaptation] 0, −1]) adapted categories. As a post-hoc analysis we also fitted the models with a phylogenetic linear mixed model for locally and globally adapted species separately, to account for phylogenetic non-independence across species because we previously found a phylogenetic signal in the degree of local adaptation[8] (Supplementary Table 4).

We also fitted linear mixed models to test whether population trends (i.e., abundances over time) were more stable for locally than for globally adapted species, and test the relation between the trends and the species bioclimatic niche. Models were fitted with population abundance (set as the continuous index of abundance $N_{jit}$) as the response variable, transformed as $\log(N_{jit} + (N_{jit}^2 + 1)^{1/2})$ to account for zero-values, and the cumulative numbers of years of observations per species in interaction with the population position within the species bioclimatic niche as explanatory variables. We used the transformation $\log(N_{jit} + (N_{jit}^2 + 1)^{1/2})$ rather than direct logarithmic as it performs similarly to a logarithmic transformation but zero remains defined avoiding the need to remove these data or add an

arbitrary value[53]. The cumulative numbers of years set per species (nested random effects) and site were set as random effects and the error as normally distributed:

$$\log\left(N_{jit} + (N_{jit}^2 + 1)^{1/2}\right) = Z_{jt} x B_{ji} + \varepsilon, \qquad (8)$$

where $N_{ji}$ is the annual abundance index of the species $j$ at the site $i$ and time $t$, $Z_{jt}$ is the cumulative number of years of observations of the species $j$ at the time $t$, and $B_{ji}$ is the bioclimatic position of the population of species $j$ at the site $i$.

To increase robustness of all our analyses we also performed separate models with alternative stricter definitions for local and global adaptation of species and without potential outliers (Supplementary Fig. 71). For locally adapted species, we modelled population change and abundance trends for those species with degree of local adaptation constrained to [0.025, 1] and without *Parnassius apollo*, and for globally adapted species from [−1, −0.025] and without *Laeosopis roboris* and *Cupido Osiris*. This led to four models for each species category (local and global; Supplementary Table 2 and Supplementary Table 5). We did not model population change using more conservative values of the degree of local adaptation because a more conservative approach reduced the number of species to <10.

We did a stepwise model reduction for interaction or additive effect between the climatic anomalies and the population position within the species bioclimatic niche. We also did model reduction between quadratic and linear response, using the full model to account for potential broadly linear responses of globally adapted species (all AIC model values provided in Supplementary Tables 6 and 7). We conducted all analyses in R 3.6.1 (R Core Team, 2019), using the package lme4[54] to fit our models and the package MuMin for model averaging[55].

### Reporting summary

Further information on research design is available in the Nature Portfolio Reporting Summary linked to this article.

### Data availability

The datasets used for this study are available from the European Butterfly Monitor Scheme via a signed license agreement (https://butterfly-monitoring.net/). Climatic data are available via ECAD website (https://www.ecad.eu/). The dataset generated for the analyses of the study is available via Zenodo [https://doi.org/10.5281/zenodo.15012265][56]. The shapefile of Europe for the Fig. 1 was obtained from the free vector and raster map data Natural Earth (https://www.naturalearthdata.com/), and the average annual temperatures from ECAD.

### Code availability

The R scripts for our analyses are also available via Zenodo [https://doi.org/10.5281/zenodo.15065537] and as a GitHub repository at ymelero/ClimaticResponses_Adapations_and_RangePosition. The use of data included within the code is licenced as Attribution-NonCommercial 4.0 International.

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

## Acknowledgements

We thank the European Butterfly Monitoring Scheme (eBMS, https://butterfly-monitoring.net/) for the data. This work and Y.M. were supported by Marie Skłodowska Curie H2020-MSCA-IF-2017 795890 project EXTINCT of the European Commission. The work was also initially supported by Foundation for Biodiversity Research and CESAB (Centre for the Synthesis and Analysis of Biodiversity, France) project LOLA. Research time of LE & TO was supported by grant NE/V007165/1. The Finnish BMS is organized and funded by the Finnish Environment Institute (Syke) and the Finnish Ministry of Environment. The UK BMS is organized and funded by Butterfly Conservation (BC), the UK Centre for Ecology and Hydrology (UKCEH), the British Trust for Ornithology (BTO) and the Joint Nature Conservation Committee (JNCC). The Catalan Butterfly Monitoring Scheme (CBMS) is funded by the Departament d'Acció Climàtica, Alimentació i Agenda Rural of the Generalitat de Catalunya, the Diputació de Barcelona and the Andorran Government. All three schemes and the authors are indebted to all volunteers who contribute data to their schemes. We thank the reviewers, whose comments improved our manuscript.

## Author contributions

L.E., M.K., D.R., R.S., C.S., T.O., and Y.M. contextualized the study and its hypotheses. Y.M. led the paper, and performed the analyses with L.E. supervised by T.O. M.K., D.R., R.S., C.S. commented on the manuscript, giving the final approval for publication.

## Competing interests

The authors declare no competing interests.
