## [Transparent Peer Review file · Communications Biology]

Species responses to weather anomalies depend on local adaptation and range position

Corresponding Author: Dr Yolanda Melero

Version 0:

Reviewer comments:

Reviewer #1

(Remarks to the Author)

The manuscript by Melero and colleagues is a valuable contribution to understanding how species may respond to climatic variations throughout their range as well as in peripheral conditions at leading or trailing edges. This is of great relevance in the face of climate change scenarios to forecast how species populations will oscillate under such changing conditions. In this work, the authors use data from butterflies from localities with a wide latitudinal gradient in western Europe for which they have well typified whether they are susceptible to changing conditions throughout their range (range/globally adapted) or whether they vary locally at some extreme of the distribution (locally adapted). This definition of "adaptation" is adopted from previous work and is the starting point for their proposal in this manuscript.

The work is based on the center-periphery hypothesis, in which populations are expected to be more abundant towards the center of the distribution range and not in the periphery. Their findings suggest that climate anomalies will negatively affect population change in locally adapted species (with more pronounced changes at the leading edge), while climate anomalies positively or negatively affect globally adapted species.

While the work is very encouraging and important, I believe it would benefit greatly from improving some definitions that are fundamental for a better understanding and traceability of its approach in a broader context and of the references to which, although they refer, the use of the terms is not always explicit. In this sense, the first thing to do is to see if they can take up the definitions of what populations are locally and globally adapted to better contextualize this work (e.g., from L48). In my opinion, the definitions are not necessarily clear and therefore it is difficult to understand the relevance of the analysis. Another point that seems critical to me related to the above is the use of the terms "ecological niche," "bioclimatic range." While they are revered and associated with Hutchinson's original approach in his definition of ecological niche, the treatment thereafter is "bioclimatic range" and this is too confusing to the reader. That is, it is not clear to them whether we are always talking about the Grinnellian scenopoetic niche based on bioclimatic cover or it may be something different. How stable this model or definition of the niche is for each species is something that must be confirmed from the beginning and in the work methods. This potential confusion permeates into the Results and discussion section where the authors explain the population change, degree of adaptation and range position (e.g., L165-168): are the authors talking about the ecological niche centroid when referring to the centre of the species bioclimatic range? And then this is even more confusing when the authors (L167) suggest that their results support the abundant niche centre hypothesis. I.e., this hypothesis was originally proposed in geographic space (not environmental space), but here they seem to mix both spaces thus creating more confusion into whether the abundant-centre hypothesis comes from the perspective of the species' geographic distribution or is better explained from/by the environmental space. Given the overall confusion with these ideas is that it is convenient to improve the explanation and clarify throughout the manuscript what are the key or relevant spaces for this explanation (e.g., see L169-170, "central niche," "average local climatic regime," "centre of the species range"). I suggest it is worth spending some time improving and homogenizing the terminology once the exploration between geographic and environmental spaces is conclusive and what processes (demographically speaking) occur or are better explained in each of these two spaces. Also, this kind of phrase could also be confused with the fundamental niche or the realized niche, is this the case here?

L52-54/L67-83. Another point that seems necessary to clarify is the relationship that exists between the geographical space from which the environmental data of the populations are being sampled and obtained and the environmental space (properly the space of the niche). In this sense, the work could improve a lot if it explains, on the one hand, that in geography it is feasible that there are the same environmental combinations that favor (or not) the populations (i.e., several times there could be a pixel/coordinate with values close to the niche centroid = site of maximum optimality/suitability). This is also

related to what is commented in L76-77: "We also expected the population changes of locally adapted species to be independent of the position within the species range." I think this may reflect the fact of several environmental optima across geographic space, for example. I suggest the authors should say why this may be the case. This could be possibly shown/explored via a correlation between distances from the geographic range center against distances from the ecological niche centroid. Are these two distances correlated in the case of these 34 butterfly species?

L52-53. The phrase ... "so the mean conditions are near the threshold of the species tolerances (e.g., minima or maxima of thermal performance)." can be very daunting given that it is possible that this is not always the case; i.e., the mean is often the place in environmental space that corresponds to the optimum, and thus it is not clear how this idea fits with the work if it is combined here with a minimum and a maximum value of performance (i.e., the limits beyond which a species could not survive). This is also related to what is expressed in L76-83.

L365-367. Bioclimatic range construction. They suggest using the abiotic niche concept but in the "range" in which the species can persist. This suggests that the ideas are mixed regarding geographic (distribution) and environmental (niche) spaces. Please, try to modify and stick to the most commonly accepted terminology accordingly.

I hope the above comments are useful to the authors in enhancing the readability and comprehension of this excellent approach. As previously mentioned, I believe the analysis is highly significant but could be further improved to make it more accessible.

Reviewer #2

(Remarks to the Author)

Comments and reviewer recommendation for manuscript "Species responses to weather anomalies depend on local adaptation and range position" by Melero et al.

This manuscript addresses an intriguing topic: the response of species to weather anomalies in relation to local or global adaptation and their position within their climatic niche. The manuscript is very well written, and the analyses are sound. My only comment concerns the figures. In Figure 1, for example, the dots indicating locations in Spain are not very visible. Adding the contour of Europe to the figure would help, as the white areas are difficult to distinguish from the sea. I also suggest labeling the figures themselves (not just in the caption using a, b, c, or d) to indicate which ones refer to locally and globally adapted species. Additionally, in the Supplementary Material, the species are currently ordered alphabetically, but I suggest grouping them by local and global adaptation: first list all 21 locally adapted species, followed by all 13 globally adapted ones.

A very minor remark: on line 202, could you restate the second hypothesis?

In the section 'Implications for forecasting and conservation,' I was hoping to read more concrete suggestions for management and/or policy measures based on the study's results. How can we buffer against anomalies? For example, through mowing, grazing, or creating more variation in vegetation structure

A very nice manuscript!

Reviewer #3

(Remarks to the Author)

This manuscript is based on a series of sound statistical analyses of the combination of similar datasets of butterfly monitoring data (i.e. standardized transect counts) in a few European countries, which together represent a latitudinal gradient (Spain, UK, Finland). In total, the analysis relies on data for 34 butterfly species of varying ecological profile. The study addresses the impact of weather anomalies. The manuscript is well written and it is clearly presented and illustrated; and the analyses are soundly executed, I believe. However, in order to evaluate whether this work is convincing and original, I have a major issue with the assumptions behind one of the key approaches of their analytical method, i.e. the comparison of what the authors refer to as locally adapted species versus globally adapted species. I have read the manuscript a couple of times, and also consulted the published paper on which this contrast of both groups (but in a larger sample of butterfly species) was previously introduced (i.e. Melero et al. 2022. *Comm. Biol.* 5: 143). I still do not grasp in a biological sense what the authors want to claim with 'locally' vs 'globally adapted species'. I thought at first that it was about semantics and that it would refer to locally adapted specialists vs more phenotypic plastic generalists for habitat use, life style or particular thermal traits, but analyzing the species list and the assigned categories of supplementary figure 35, I was completely lost. For me it hence completely unclear what the groups represent biologically. This is however an essential ingredient in order to understand and so decently review this manuscript. I even got more confused when reading the sentence on L. 418: 'To increase robustness of all our analyses we also performed separate models with alternative stricter definitions for local and global adaptations of species'. For me this conceptual (or semantic) ambiguity needs to be addressed before the merit of the current contribution can be really assessed.

Moreover, in their previous paper, one of the main conclusions is the existence of phylogenetically signatures in the effects of climate anomalies on population trends of butterfly species. However, unless I missed some relevant information, the phylogenetic relationships between the tested species in the current manuscript have not been taken into account. This does not seem to be coherent, and at least, it would require some justification as some species belong to the same genus.

I would argue that the authors need to address these issues before I could really assess the scientific value of their work.

Version 1:

Reviewer comments:

Reviewer #1

(Remarks to the Author)

The work by Melero et al., in its latest version, successfully incorporated and enhanced several of the suggestions proposed by the reviewers during the last round of revisions. I believe the explanations have improved and are now clearer while maintaining simplicity. However, I have a few observations that I think could help refine some points mentioned in the "Main" section of the manuscript.

In lines 47–52, the authors state:

“Hence, populations at the distribution range centre are closer to the centre of the bioclimatic niche and more stable and abundant as a consequence (i.e. the ‘abundant niche centre hypothesis’), while populations at the distribution margins are nearer to the thresholds of the species tolerances (e.g. minima or maxima of thermal performance).”

However, I believe this assertion reiterates a pattern similar to that proposed by the abundant-centre hypothesis. While the hypothesis is valid in its premise and original formulation, I suggest the authors clarify that this pattern is not always to be expected. There could be multiple coordinates in geographic space with niche suitability values close to the optimum. This pattern would only be expected if there is a strong correlation between the distances to the niche centroid and the distances to the geographic range center, which was a suggestion to check in the last round of revisions. I recommend revising and refining this statement to leave open the alternative (e.g., see Figure 1 of Lira-Noriega & Manthey 2014 Evolution).

This idea also appears in lines 178–181. I would question again whether the pattern observed for locally versus globally adapted species might be an artifact of the measurement approach. For instance, in the case of locally adapted species, did you only use local rather than global measurements? I suggest these ideas should be further developed, addressing the proposed alternative solutions and clarifying whether this outcome is potentially due to correlations (or lack thereof) between ecological and geographic spaces or the species selected for analysis.

Minor comments:

Line 123: At the end of the sentence, there is the word ‘fin’, which appears to be an error. Please check and correct it.

Line 184: Instead of “bioclimatic/ecological niche,” consider revising to “ecological (bioclimatic) niche.”

Line 184: Replace “climatic” with “climate” for consistency.

Reviewer #2

(Remarks to the Author)

Comments and reviewer recommendation for the revised manuscript “Species responses to weather anomalies depend on local adaptation and range position” by Melero et al.

The authors did a thorough job with this revision and the manuscript certainly improved thanks to the reviewers. But, I still have two remarks on this revision:

1. In Figure 1, the dots are still not clearly visible and might cause problems as well for colourblind people. I would recommend finding a different combination of colours to make this important graph more ‘readable’.
2. In the Supplementary Material, Fig. 35 could be rearranged by putting the Locally adapted species together first, followed by the globally adapted ones as was done in Supplementary Table 1.

Apart from that, I have no further comments

Great work!

Reviewer #3

(Remarks to the Author)

Previously, I had identified a couple of issues that mainly related to i) the concept/definition of locally vs globally adapted species, and ii) phylogenetical controls. In the revised version, the authors anticipated/corrected the manuscript in a convincing way (or added analyses). Hence, my initial reservation has disappeared after reading carefully this reworked version of the manuscript. Moreover, I believe the authors have also done a good job responding adequately to the comments of the other reviewers, at least as far as I can see. I am happy with this manuscript as it stands now. It will be a significant contribution to the field.

Version 2:

Reviewer comments:

Reviewer #1

(Remarks to the Author)

I appreciate the effort made by the authors in completing the final round of revisions. I suggest that they include a color

legend in the last generated figure (Supplementary Figure 74) for consistency with other figures, making it easier to interpret (e.g., "Colours indicate locally adapted species in blue and globally adapted species in orange."). Additionally, it would be helpful to specify in the figure legend how the significance of the relationships was evaluated.

Reviewer #2

(Remarks to the Author)

The authors have done a good job with my final comments and I have no further comments. Authors should carefully check species names though (e.g. in the rebuttal 'Argynnis adippe' is written incorrectly as 'Arginis adipe').

Response to Reviewers' comments – COMMSBIO-24-3403-T

Reviewers' comments:

Reviewer #1 (Remarks to the Author):

We are grateful for the reviewer's positive and thoughtful comments, which have significantly helped us improve the clarity and quality of our manuscript. We have implemented most of the recommendations and or provided explanations for retaining aspects of the original approach. Please, see the response to each comment detailed below.

The manuscript by Melero and colleagues is a valuable contribution to understanding how species may respond to climatic variations throughout their range as well as in peripheral conditions at leading or trailing edges. This is of great relevance in the face of climate change scenarios to forecast how species populations will oscillate under such changing conditions.

In this work, the authors use data from butterflies from localities with a wide latitudinal gradient in western Europe for which they have well typified whether they are susceptible to changing conditions throughout their range (range/globally adapted) or whether they vary locally at some extreme of the distribution (locally adapted). This definition of "adaptation" is adopted from previous work and is the starting point for their proposal in this manuscript.

The work is based on the center-periphery hypothesis, in which populations are expected to be more abundant towards the center of the distribution range and not in the periphery. Their findings suggest that climate anomalies will negatively affect population change in locally adapted species (with more pronounced changes at the leading edge), while climate anomalies positively or negatively affect globally adapted species.

While the work is very encouraging and important, I believe it would benefit greatly from **improving some definitions that are fundamental for a better understanding and traceability of its approach in a broader context and of the references to which, although they refer, the use of the terms is not always explicit.**

1. In this sense, the first thing to do is to see if they **can take up the definitions of what populations are locally and globally adapted** to better contextualize this work (e.g., from L48). In my opinion, the definitions are not necessarily clear and therefore it is difficult to understand the relevance of the analysis.

We have now explain defined what we mean by locally and globally adapted species (lines 61-70), and added a glossary including their definition alongside other terms including degree of local adaptation, distributional range and bioclimatic niche. Please also be aware we have re-organized our introduction to expose better these concepts.

2. Another point that seems critical to me related to the above is **the use of the terms "ecological niche," "bioclimatic range."** While they are revered and associated with Hutchinson's original approach in his definition of ecological niche, the treatment thereafter is "bioclimatic range" and this is too confusing to the reader. That is, it is not clear to them whether we are always talking about the Grinnellian scenopoetic niche based on bioclimatic cover or it may be something different. **How stable this model or definition of the niche is for each species is something that must be confirmed from the beginning and in the work methods.** This potential confusion permeates into the Results and discussion section where the authors explain the population change, degree of adaptation and range position (e.g., L165-168): are the authors talking about the ecological niche centroid when referring to the centre of the species bioclimatic range? And then this is even more confusing when the authors (L167) suggest that their results support the **abundant niche centre hypothesis. I.e., this hypothesis was originally proposed in geographic space (not environmental space), but here they seem to mix both spaces thus creating more confusion into whether the abundant-centre hypothesis comes from the perspective of the species' geographic distribution or is better explained from/by the environmental space.** Given the overall confusion with these ideas is that it is convenient to improve the explanation and clarify throughout the manuscript what are the key or relevant spaces for this explanation (e.g., see L169-170, "central niche," "average local climatic regime," "centre of the species range"). I suggest it is worth spending some time improving and homogenizing the terminology once the exploration between geographic and environmental spaces is conclusive and what processes (demographically speaking) occur or are better explained in each of these two spaces. **Also, this kind of phrase could also be confused with the fundamental niche or the realized niche, is this the case here?**

We agree that our use of the geographical range and ecological niche (specifically the Grinnellian niche, based on climatic data in our case) may have caused some confusion by mixing the concepts of geographical range and ecological niche.

From our perspective they are related concepts. The abundant centre hypothesis posits that species will be most abundant at their range centre (in geographic space) and populations at the range edges will be most variable. Support for this hypothesis has historically been mixed, and this is assumed to be because the distribution of species/populations corresponds not perfectly to geography (i.e. by latitude and longitude) per se but by environmental variables. So, for example, a population of cold-associated species in Europe could be highly abundant in the southern parts of their range where there are mountains. Hence, the abundant-niche hypothesis is a refinement of the abundant centre hypothesis positing that species will be most abundant at the niche centre (Martínez-Meyer et al. 2013). From our understanding to date, this hypothesis is better supported (e.g. for birds, Osorio-Olvera 2020; Osorio-Olvera et al. 2019, and ectotherms Mills et al. 2017; Oliver et al. 2012; but see Pironon et al. 2017). Therefore, we situate our research in the context of the abundant-niche centre hypothesis and reference the abundant centre hypothesis due to the connection between the two ideas. We measure this position in the range within environmental space (based on species-specific temperature or precipitation variables), that is what we refer to as bioclimatic range position, and therefore refer to it as the centre of a species' bioclimatic niche'. Note, this bioclimatic niche may also implicitly capture biotic factors such as host plants and competition that constrain the distribution of a species. So, it is a measure of the *realised niche*. For example, a population of species may be limited to colder, mountain locations due to the presence of a specialist host plant. In the example above, the mountain populations in the South of Europe may actually be in the core of the bioclimatic range and expected to be more stable and abundant.

In reality, species distributions and abundances do not always correspond even with this improved measure of niche space, because there may be biases in the measured distribution of species. European monitoring data are fairly complete at the scale we are considering distribution, however there are several species whose range extends beyond Europe (*Leptidea sinapsis*, *Melanargia galathea*, *Vanessa cardui*, *Vanessa Atalanta*; Supplementary Table 1), so the extremes of their climatic range are not fully captured and we might slightly over- or underestimate their niche centroid. We discuss this limitation in the manuscript.

We have re-written part of the introduction explaining our definitions and the relationships between these concepts. We have also changed bioclimatic range to bioclimatic niche throughout, i.e. we explicitly refer to either distributional range or bioclimatic niche. Finally, we have also added definitions of key terms to the Glossary.

3. L52-54/L67-83. Another point that seems necessary to clarify is the relationship that exists between the geographical space from which the environmental data of the populations are being sampled and obtained and the environmental space (properly the space of the niche). In this sense, the work could improve a lot if it explains, on the one hand, that in geography it is feasible that there are the same environmental combinations that favor (or not) the populations (i.e., several times there could be a pixel/coordinate with values close to the niche centroid = site of maximum optimality/suitability).

Yes, we agree. As commented above we have now explained the relation between geographical and environmental spaces further and modified the text to clarify our approach, in a way which we believe addresses this concern.

This is also related to what is commented in L76-77: "We also expected the population changes of locally adapted species to be independent of the position within the species range." I think this may reflect the fact of several environmental optima across geographic space, for example. I suggest the authors should say why this may be the case.

The reviewer is right in stating that there can be several environmental optima across geographic space, such as the example of some high altitudes in Spain being colder than some regions in the UK. Yet, we avoided these potential cases of decoupling between geographical and environmental spaces by using bioclimatic niche approach. Our sentence is now corrected to state "We also expected the population changes of locally adapted species to be independent of the position within the species bioclimatic niche".

This could be possibly shown/explored via a correlation between distances from the geographic range center against distances from the ecological niche centroid. Are these two distances correlated in the case of these 34 butterfly species?

While this is an excellent idea we lack of the full geographical data to do so. However, as commented above, there are several previous studies relating geographical range and climatic niche for butterfly species (including our 34, e.g. (Mills et al. 2017).

4. L52-53. The phrase ...“so the mean conditions are near the threshold of the species tolerances (e.g., minima or maxima of thermal performance).” can be very daunting given that it is possible that this is not always the case; i.e., the mean is often the place in environmental space that corresponds to the optimum, and thus it is not clear how this idea fits with the work if it is combined here with a minimum and a maximum value of performance (i.e., the limits beyond which a species could not survive). This is also related to what is expressed in L76-83.

The sentence relates to the ‘abundant niche center hypothesis’, hence the species tolerance referred to here relates the bioclimatic niche space. This should be now clear with the changes we have made in the introduction following the reviewer's advice.

5. L365-367. Bioclimatic range construction. **They suggest using the abiotic niche concept but in the “range” in which the species can persist. This suggests that the ideas are mixed regarding geographic (distribution) and environmental (niche) spaces.** Please, try to modify and stick to the most commonly accepted terminology accordingly.

As above, we are indeed using environmental (bioclimatic) niche space throughout and clarify this in the new text. Note, also that our measure of environmental niche space may also implicitly include biotic constraints (see response to first comment).

I hope the above comments are useful to the authors in enhancing the readability and comprehension of this excellent approach. As previously mentioned, I believe the analysis is highly significant but could be further improved to make it more accessible.

We would like to remark that the reviewer's comments were very useful and have helped us to improve the manuscript. We thank the reviewer for the insights.

Reviewer #2 (Remarks to the Author):

Comments and reviewer recommendation for manuscript “Species responses to weather anomalies depend on local adaptation and range position” by Melero et al.

This manuscript addresses an intriguing topic: the response of species to weather anomalies in relation to local or global adaptation and their position within their climatic niche. The manuscript is very well written, and the analyses are sound. My only comment concerns the figures.

We are grateful for the positive comments of the reviewer which are very encouraging. We have carried out most of the recommendations and argued those that we decided not to follow. Please, see the response to each comment bulleted below.

1. In Figure 1, for example, the dots indicating locations in Spain are not very visible. Adding the contour of Europe to the figure would help, as the white areas are difficult to distinguish from the sea.

We agree points and light colored areas were not visible enough and we have changed them.

2. I also suggest labelling the figures themselves (not just in the caption using a, b, c, or d) to indicate which ones refer to locally and globally adapted species.

Done. Please be aware that we have slightly modified our terminology based on other reviewers' comments. We have not changed the Supplementary Figures as they are color-coded, but we have added the labels in Supplementary Figures 36 and 71.

3. Additionally, in the Supplementary Material, the species are currently ordered alphabetically, but I suggest grouping them by local and global adaptation: first list all 21 locally adapted species, followed by all 13 globally adapted ones.

Thanks, this is now amended in the figures and Table 1 in the Supplementary Material.

4. A very minor remark: on line 202, could you restate the second hypothesis?

Done, this now read in lines 216-219.

5. In the section ‘Implications for forecasting and conservation,’ I was hoping to read more

concrete suggestions for management and/or policy measures based on the study's results. How can we buffer against anomalies? For example, through mowing, grazing, or creating more variation in vegetation structure.

We are aware that the management approaches presented at the end of our discussion are general and focus on contextualized strategies, considering whether species are locally or globally adapted and their range position. We have included some examples to illustrate potential directions for management strategies. While we provide some examples of actions, we prefer not to develop this further, because we have not analyzed the specific effects of management.

A very nice manuscript!
Thank you for your positive and encouraging words!

Reviewer #3 (Remarks to the Author):

This manuscript is based on a series of sound statistical analyses of the combination of similar datasets of butterfly monitoring data (i.e. standardized transect counts) in a few European countries, which together represent a latitudinal gradient (Spain, UK, Finland). In total, the analysis relies on data for 34 butterfly species of varying ecological profile. The study addresses the impact of weather anomalies. The manuscript is well written and it is clearly presented and illustrated; and the analyses are soundly executed, I believe.

However, in order to evaluate whether this work is convincing and original, I have a major issue with the assumptions behind one of the key approaches of their analytical method, i.e. the comparison of what the authors refer **to as locally adapted species versus globally adapted species**. I have read the manuscript a couple of times, and also consulted the published paper on which this contrast of both groups (but in a larger sample of butterfly species) was previously introduced (i.e. Meleró et al. 2022. *Comm. Biol.* 5: 143). I still do not grasp in a biological sense what the authors want to claim with 'locally' vs 'globally adapted species'.

Thank you for the positive words on the robustness of our statistical analysis. We have now provided clearer definitions for locally and globally (lines 61-70). Additionally, we have included a glossary that contains their definitions alongside other terms, such as degree of local adaptation, distributional range, and bioclimatic niche. Overall, we focus here on a measure of physiological adaptation assessed using a simple method of whether species' populations have a performance optimum centred on the local conditions or the mean conditions of the entire distribution. Importantly this metric allowed us to make specific predictions about how the populations will respond to anomalies at range edges as species with performance centered on the global optima should have contrasting responses between range edges, while those centred on the local optima should have similar responses across the range. This prediction was supported in our results.

I thought at first that it was about semantics and that it would refer to locally adapted specialists vs more phenotypic plastic generalists for habitat use, life style or particular thermal traits, but analyzing the species list and the assigned categories of supplementary figure 35, I was completely lost. For me it hence completely unclear what the groups represent biologically. This is however an essential ingredient in order to understand and so decently review this manuscript. I even got more confused when reading the sentence on L. 418: 'To increase robustness of all our analyses we also performed separate models with alternative stricter definitions for local and global adaptations of species'. For me this conceptual (or semantic) ambiguity needs to be addressed before the merit of the current contribution can be really assessed.

We understand this confusion, indeed being locally adapted to climate conditions could arise from several processes resulting from genetic, morphological/phenotypic to behavioral variation. As we detail in the new glossary, we are specifically referring to physiological local adaptation as measured by our approach of assessing responsiveness to local or global climate anomalies. This indicates where a species population's thermal (or other key variable aridity/precipitation) performance optimum is centred on the local conditions or the mean conditions across the entire range.

Analysing the ability of the species to have adapted populations (either by plasticity or by evolutionary selection) using trait values is certainly interesting, i.e. exploring the mechanisms underlying local adaptation to climatic conditions, but such data are not available at the individual or population level across all these species. However, the ability of a species to have adapted populations (either by plasticity or by evolutionary selection) is captured with our measure degree of local adaptation.

Moreover, in their previous paper, one of the main conclusions is the existence of phylogenetically signatures in the effects of climate anomalies on population trends of butterfly species. However, unless I missed some relevant information, the phylogenetic relationships between the tested species in the current manuscript have not been taken into account. This does not seem to be coherent, and at least, it would require some justification as some species belong to the same genus.

Yes, we previously found a strong phylogenetic signal in the degree of local adaptation of the species, i.e., on the ability of a given species' populations to adapt to the local conditions (see glossary for more detailed definition). We take the reviewers point that a phylogenetic analysis would allow us to be more confident that the different patterns we see for locally versus globally adapted species are due to this attribute and not just because of different genera/families that are patterned between these two groups. Therefore, we have extended our statistical analysis by fitting mixed effects models with phylogenetic relationship as a random effect. The results are qualitatively similar and our conclusions unchanged and we include these new results in supplementary material and refer to them in the main text.

The additional text can be found at lines 211-213 and 426-429 of the main manuscript, with results in table format in Supplementary Material Table 4.

I would argue that the authors need to address these issues before I could really assess the scientific value of their work.

Hopefully the clarifications and additions of the glossary we have provided will aid your assessment.

Reviewers' comments:

We thank again the three reviewers for their helpful comments and advice. Please see below for detailed responses.

Reviewer #1 (Remarks to the Author):

The work by Melero et al., in its latest version, successfully incorporated and enhanced several of the suggestions proposed by the reviewers during the last round of revisions. I believe the explanations have improved and are now clearer while maintaining simplicity. However, I have a few observations that I think could help refine some points mentioned in the "Main" section of the manuscript.

In lines 47–52, the authors state:

"Hence, populations at the distribution range centre are closer to the centre of the bioclimatic niche and more stable and abundant as a consequence (i.e. the 'abundant niche centre hypothesis'), while populations at the distribution margins are nearer to the thresholds of the species tolerances (e.g. minima or maxima of thermal performance)."

However, I believe this assertion reiterates a pattern similar to that proposed by the abundant-centre hypothesis. While the hypothesis is valid in its premise and original formulation, I suggest the authors clarify that this pattern is not always to be expected. There could be multiple coordinates in geographic space with niche suitability values close to the optimum.

A: We agree with the reviewer, for that reason we previously added the statement in lines 44-47: "This pattern is expected because, although some environmental (e.g. altitudinal differences (Loarie et al. 2009)) and methodological (Dallas et al. 2017; Santini et al. 2019) nuances apply, there is a general concordance between the geographical/distributional range space and the species' optimal environmental space (the bioclimatic/ecological niche, but see (Pironon et al. 2015; Martínez-Meyer et al. 2013; Osorio-Olvera 2020; Osorio-Olvera et al. 2019; Pironon et al. 2017))." With this statement we acknowledged that the link between the geographical/distributional range and the climatic niche (and consequently, the abundant center hypothesis and its connection with the abundant niche center hypothesis) does not always align.

We also acknowledged this in the definition of the Bioclimatic Niche within the new added Glossary, where we commented that evidence suggest a correlation between the bioclimatic niche and the distributional range for thermal performance in ectotherms but some factors may modified this relationship: "For ectotherms, the bioclimatic niche aligns closely with their distributional range due to the significant influence of temperature on their physiological processes¹⁻³, although it can be different due to factors such as altitude."

This pattern would only be expected if there is a strong correlation between the distances to the niche centroid and the distances to the geographic range center, which was a suggestion to check in the last round of revisions. I recommend revising and refining this statement to leave open the alternative (e.g., see Figure 1 of Lira-Noriega & Manthey 2014 Evolution).

A: We have now done a correlation analyses between the bioclimatic nice position of the species populations and the range position, the latter calculated as the relative range position (RRP) across a latitudinal vector (rather than across the geographic center since we lack of this data) following the approach in Mills *et al* (<https://doi.org/10.1111/geb.12659>). This does not directly relates to the center of the range position since we lack of the abundance data of the species entire distribution, but uses the relative position of the population within a latitudinal vector of the species distribution. Hence, negative correlations should be observed if bioclimatic niche position properly aligns with geographical range distributions. RRP ranges 0-1 from lower to higher latitudes. Our correlation results, even with using this proxy, captures the alignment of both variables (RRP and bioclimatic niche). Please check lines 421-433 of Methods in the MS, and Supplementary Figure 74 (also pasted below):

Supplementary Figure 74. Pearson correlation between the population position within the bioclimatic niche and the relative range position of each species. Bioclimatic range position ranges [-1, 1] from leading (position -1) to trailing margins (position 1). Relative range position ranges [0, 1] from lower latitudes (trailing margins) to higher latitudes (leading margins). Negative correlations were expected. All species failing at correlating margins had a degree of lower adaptation $< |0.025|$, hence they were not included in the most conservative models (Supplementary Figs. 36 and 71), except for *Argynnis adippe* ($d_{la} = 0.033$, a locally adapted species to precipitation, present in high altitudinal sites in Spain) and *Lysandra coridon* ($d_{la} = 0.066$, a locally adapted best adapted to temperature, absent in the UK). All correlations were significant.

This idea also appears in lines 178–181. I would question again whether the pattern observed for locally versus globally adapted species might be an artifact of the measurement approach. For instance, in the case of locally adapted species, did you only use local rather than global measurements?

A: Yes, we used local performance curves (i.e., populations, site-specific) in relation to local climatic anomalies to analyze the patterns across the species bioclimatic niche, following our H1. Further, local and global adapted species are defined based on their strength of their response to the local versus global climatic anomalies. Both: the use of local climatic anomalies data for the analyses and, the use for the species classification are stated throughout the MS and the new Glossary:

- MS lines 62-70 we introduced the terms "Locally" and "Globally" adapted species. Referring to the Glossary for their definition: "Degree of local adaptation:...Our score of local adaptation is calculated based on the difference in explanatory power (R^2) of models using local versus global climatic anomalies, indicating the relative influence of local climate on population dynamics..." Likewise: "Locally Adapted Species: Species whose population dynamics are more strongly influenced by climatic anomalies at a local scale (e.g., deviations from average conditions at specific sites)...."

- MS lines 82-85: the scale of the climatic anomalies is also specified: "Specifically, we predict that population growth rates of locally adapted species follow a non-linear quadratic response to local climatic anomalies, with a maximum performance around the local average conditions (i.e. when no anomalies occur) decreasing below and above them (i.e. when local anomalies occur)...."

- MS lines 87: Likewise "for globally adapted species we predicted population responses to the local climatic anomalies to differ"
- MS lines Fig. 1: local anomalies was commented: "Predicted population responses of globally and locally adapted species to local climatic anomalies in relation to population niche position". Likewise in Fig 2: " Fig. 2. Population change in relation to the local climatic anomalies of the year..."
- MS lines 145-149 (Results). "We found species population sensitivity to the local climatic anomalies varied both depending on whether they were locally or globally adapted species and, for the latter, on the position of the population within the species bioclimatic niche..."

The mention to local measures appear in the rest of the main MS and methods, and in SM. We have now added "local" to this sentence:

- MS line 139: "To test our first hypothesis, we modelled how annual population change varied in relation to the local climatic anomalies in interaction with the bioclimatic position"

I suggest these ideas should be further developed, addressing the proposed alternative solutions and clarifying whether this outcome is potentially due to correlations (or lack thereof) between ecological and geographic spaces or the species selected for analysis.

A: In line with our responses above, we believe we now show that both geographical and ecological (bioclimatic niche) aligns in the study system.

Minor comments:

Line 123: At the end of the sentence, there is the word 'fin', which appears to be an error. Please check and correct it.

A: Thanks. This has been corrected, "fin" was corrected to "in"

Line 184: Instead of "bioclimatic/ecological niche," consider revising to "ecological (bioclimatic) niche."

A: This has been done now

Line 184: Replace "climatic" with "climate" for consistency.

A: This has been done now

Reviewer #2 (Remarks to the Author):

Comments and reviewer recommendation for the revised manuscript "Species responses to weather anomalies depend on local adaptation and range position" by Melero et al.

The authors did a thorough job with this revision and the manuscript certainly improved thanks to the reviewers. But, I still have two remarks on this revision:

1. In Figure 1, the dots are still not clearly visible and might cause problems as well for colourblind people. I would recommend finding a different combination of colours to make this important graph more 'readable'.

A: Thanks. This has now been done. We now use dots that are colored and shaped differently (circles, triangles, and squares) to improve visibility, and we have adopted a more user-friendly color palette. We also tested other palettes (e.g., viridis) for Figures 2-4, but the dots were not distinguishable.

2. In the Supplementary Material, Fig. 35 could be rearranged by putting the Locally adapted species together first, followed by the globally adapted ones as was done in Supplementary Table 1.

A: This has been done now

Apart from that, I have no further comments

Great work!

A: Thank you for your feedback and advises

Reviewer #3 (Remarks to the Author):

Previously, I had identified a couple of issues that mainly related to i) the concept/definition of locally vs globally adapted species, and ii) phylogenetical controls. In the revised version, the authors anticipated/corrected the manuscript in a convincing way (or added analyses). Hence, my initial reservation has disappeared after reading carefully this reworked version of the manuscript. Moreover, I believe the authors have also done a good job responding adequately to the comments of the other reviewers, at least as far as I can see. I am happy with this manuscript as it stands now. It will be a significant contribution to the field.

A: Thank you for your revision that has helped to improve the MS.

We thank again the reviewers for their effort checking the manuscript and for their advises, which have improved our manuscript.

Please see below for detailed responses.

I appreciate the effort made by the authors in completing the final round of revisions. I suggest that they include a color legend in the last generated figure (Supplementary Figure 74) for consistency with other figures, making it easier to interpret (e.g., "Colours indicate locally adapted species in blue and globally adapted species in orange."). Additionally, it would be helpful to specify in the figure legend how the significance of the relationships was evaluated.

We have now included the sentence indicating the color legend as recommended. In addition, we have also changed the order of the plots in the Figure be consistent with the previous figures, showing first the locally adapted species and then the globally adapted ones. We have also added the significant level at the figure legend, and explained the meaning of correlation failure. Now the Figure is as follows:

Supplementary Figure 74. Pearson correlation between the population position within the bioclimatic niche and the relative range position of each species. Bioclimatic range position ranges [-1, 1] from leading (position -1) to trailing margins (position 1). Relative range position ranges [0, 1] from lower latitudes (trailing margins) to higher latitudes (leading margins). Negative correlations were expected. All species failing at correlating margins (correlation < 0.4), or with positive correlations ha degree of lower adaptation $< |0.025|$, hence they were not included in the most conservative models (Supplementary Figs. 36 and 71), except for *Argynnis adippe* (dla = 0.033, a locally adapted species to precipitation, present in high altitudinal sites in Spain) and *Lysandra coridon* (dla = 0.066, a locally adapted best adapted to temperature, absent in the UK). All other correlations were negative and significant (p-values < 0.005). Colours indicate locally adapted species in blue and globally adapted species in orange.

Reviewer #2 (Remarks to the Author):

The authors have done a good job with my final comments and I have no further comments. Authors should carefully check species names though (e.g. in the rebuttal 'Argynnis adippe' is written incorrectly as 'Arginis adipe').

We have now checked for mistakes, this was an error in the rebuttal letter. Our apologies.